# Analysis of Nonlinear Vibration of Functionally Graded Simply Supported Fluid-Conveying Microtubes Subjected to Transverse Excitation Loads

**DOI:** 10.3390/mi13122114

**Published:** 2022-11-30

**Authors:** Tao Ma, Anle Mu

**Affiliations:** 1School of Mechanical and Precision Instrument Engineering, Xi’an University of Technology, Xi’an 710048, China; 2School of Mechanical Engineering, Henan University of Engineering, Zhengzhou 451191, China

**Keywords:** strain gradient theory, transverse excitation loads, fluid-conveying microtubes, nonlinear vibration, functionally-graded materials

## Abstract

This paper presents a nonlinear vibration analysis of functionally graded simply supported fluid-conveying microtubes subjected to transverse excitation loads. The development of the nonlinear equation of motion is based on the Euler–Bernoulli theory, Hamilton principle and Strain gradient theory. The nonlinear equation of motion is reduced to a second-order nonlinear ordinary differential equation by the Galerkin method. The Runge–Kutta method is adapted to solve the equation, and the effects of the dimensionless microscale parameters, the amplitude and frequency of excitation loads on the stability of the microtubes system are analyzed. It is found that when the microtube diameter is equal to the material length scale parameter, the microtube movement pattern is quasi-periodic. With the increase of the dimensionless microscale parameter, the microtube movement changes from quasi-periodic to chaos. The smaller the power-law index of volume fraction, the smaller the vibration displacement of microtubes and the better the stability. The larger the amplitude of excitation loads is, the larger the vibration displacement of the microtubes will be. When the frequency of excitation loads is equal to the natural frequency of the microtubes, it will have resonance and the vibration displacement will increase significantly.

## 1. Introduction

Fluid-conveying microtubes, as the main component of microfluidic transport or storage, have been widely used in various micro/nano-electromechanical systems, such as micropumps, microvalves and micro-actuators [1]. The structure size of fluid-conveying microtubes is generally used on the micrometer or even nanometer scale. Relevant studies have proved that the length scale parameter of the materials has a significant impact on the mechanical properties of the microstructure at the microscale [2,3]. As an important part of the micro/nano-electromechanical system, the stability of the fluid-conveying microtubes is of great significance to the stable operation of the entire micro-system. Vibration isolation is an effective vibration control method, which can improve the stability of structures [4]. However, it is very difficult to use vibration isolation technology for the vibration control of micro/nanotubes.

At present, scholars have carried out a lot of research on the vibration mechanism of micro/nanotubes. Among these studies, the research results of carbon nanotubes are the most abundant. For example, Babaei et al. [5,6,7] established the nonlinear dynamic equation of functionally graded porous carbon nanotubes under thermo-mechanical loads and analyzed the effects of thermal loads, geometric imperfections and microscale parameters on system stability. For carbon nanotubes in elastic fields, Yinusa et al. [8] established the dynamic equation of carbon nanotubes embedded in nonlinear viscoelastic foundations to transport nano-magnetic fluids and investigated the nonlinear thermodynamic stability of different end-shaped single-walled carbon nanotubes under the action of magnetic fields. Based on the thermo-moisture elasticity theory and Eringen’s non-local theory, Dini et al. [9] derived the motion equations of uniformly curved double-wall carbon nanotubes in a hot and humid environment and analyzed the thermo-moisture vibration stability of the carbon nanotubes under the action of a magnetic field.

Considering the viscoelastic effect of the foundation, the carbon nanotubes for transporting magnetic nanofluids under different boundary conditions were studied by Mahmoudpour et al. [10] to explore the effect of the longitudinal magnetic fields on the stability of carbon nanotubes. Ninf et al. [11] employed the analytical approach to study the nonlinear vibration of a functionally graded sandwich shell containing heavy water in an elastic foundation under thermo-mechanical loads. As for the influence of the external magnetic fields on carbon nanotubes, Sadeghi-Goughari et al. [12] researched the microscale effects and transverse magnetic fields on the vibrational properties of cantilever carbon nanotubes. Based on the Euler–Bernoulli beam model, Tong et al. [13] analyzed the effects of external magnetic fields and temperature fields on the stability of the functionally graded cantilever nanotubes under the effect of thermo-magnetic coupling. 

By using the high-precision iterative method, Lyu et al. [14] studied the vibration properties of carbon nanotubes in multi-physics fields. Cheng et al. [15] used the dynamic stiffness method to solve the motion equation of a segmented fluid transport nanotube composed of carbon nanotubes and boron nitride nanotubes and researched the effects of aspect ratios and non-local parameters on the vibration characteristics of the hybrid nanotubes. Bahaadini et al. [16] adapted the Galerkin method to analyze the size effects, Knudsen numbers and boundary conditions on the stability of carbon nanotubes. In addition, Ninh et al. [17] applied the fourth-order Runge–Kutta method to resolve the equations of motion and studied the fundamental frequencies of the system using the time histories and bifurcation diagram. In the above studies, different methods were used to establish the motion equation of the carbon nanotubes, and the effects of parameters were studied, such as functionally graded materials, microscale parameters, geometric imperfections, magnetic fields, temperature fields and excitation loads. However, these studies are focused on the nanotubes, and do not consider tubes of the micron scale.

Micron-scale fluid-conveying microtubes are 10^3^ orders of magnitude larger than carbon nanotubes. They have been widely used as fluid-conveying and storage devices in various MEMS. In recent years, research on the dynamic stability of fluid-conveying microtubes has received extensive attention from scholars. Ghayesh et al. [18] first proposed a scale-dependent theoretical model considering curvature nonlinearity, derived the nonlinear coupled motion equation of fluid-conveying microtubes and studied the complex viscoelastic coupling dynamics of fluid-conveying microtubes. Considering the inextensibility of the fluid-conveying microtubes and the neglect of coupled stress effects, Dehrouyeh-Semnani et al. [19] used the Hamiltonian principle to establish the motion equation of the fluid-conveying microtubes. Hosseini et al. [20] established the dynamic equation of fluid-conveying microtubes by using modified strain gradient theory and Euler–Bernoulli beam model and analyzed the size-dependent stability of cantilever fluid-conveying microtubes. Based on the modified couple stress theory, Hu et al. [21] derived the motion equation of cantilever fluid-conveying microtubes by using a nonlinear theoretical model and discussed the possible size-dependent nonlinear response. For the three-dimensional nonlinear dynamic analysis, Guo et al. [22] deduced the dynamic equation of cantilever fluid-conveying microtubes based on the modified couple stress theory and Hamiltonian principle and studied the two possible types of periodic motions by means of averaging methods and numerical simulations.

For functionally graded simply supported fluid-conveying microtubes, Setoodeh et al. [23] established the functionally graded fluid-conveying microtubes nonlinear motion equation and analyzed the effects of microscale parameters and power-law index on the fundamental frequency of the microtubes and the critical fluid velocity. Under the framework of non-local strain gradient theory, She et al. [24] derived the nonlinear motion equation of the functionally graded porous microtubes, and used a two-step perturbation method to analyze the effects of microscale parameters, pore volume fractions and power-law index on the system. Regarding the vibration characteristics of functionally graded fluid-conveying microtubes under the action of thermo-magnetic coupling, Ma et al. [25] analyzed the influence of thermo-magnetic coupling parameters, microscale parameters and power-law index of gradient materials on the dynamic properties of fluid-conveying microtubes. Guo et al. [26] established the nonlinear motion equation of a cantilever microtube with symmetrical constraints, solved the equation by the Galerkin method and discussed the influence of material length scale parameters and constraints on the critical velocity of the microtubes. In addition, Sabahi et al. [27], Yuan et al. [28] and Amiri et al. [29] carried out dynamic research on the fluid-conveying microtubes from the aspect of the dynamics model, solution method and microscale parameters.

Excitation loads are a phenomenon that fluid-conveying microtubes often encounter in engineering applications. For example, the axial force generated by temperature variation and magnetic force will affect the stability of the fluid-conveying microtubes. There have been a lot of theoretical studies on the effect of excitation loads on the dynamic behavior of carbon nanotubes [30,31,32,33,34], which have laid a solid theoretical foundation for the engineering application of carbon nanotubes. However, there are few studies on micron-scale fluid-conveying microtubes under excitation loads. For instance, Zhu et al. [35] used the Euler–Bernoulli beam theory and Hamilton principle to deduce the motion equation of functionally graded fluid-conveying microtubes subjected to distributed loads and used the variational iterative method and multi-scale direct method to analyze the nonlinear free vibration and main resonance of fluid-conveying microtubes. Based on the modified couple stress theory, Dehrouyeh-Semnani et al. [36] derived the nonlinear equation of forced motion for microtubes under harmonic loads and studied the nonlinear size-dependent main resonance characteristics of microtubes in the subcritical region through the frequency response and force response.

It can be from the above studies that the research on fluid-conveying microtubes under excitation loads is still limited, especially the nonlinear research on functionally graded fluid-conveying microtubes under external excitation loads. Therefore, based on the strain gradient theory and the Hamiltonian principle, the nonlinear motion equation of functionally graded simply supported fluid-conveying microtubes is established under transverse excitation loads in this paper. Through the motion phase diagram and bifurcation diagram, the effects of functionally graded materials, microscale parameters and excitation loads on the dynamic behavior of fluid-conveying microtubes are explored.

## 2. Motion Equation

The fluid-conveying microtubes are placed horizontally. Regardless of the effect of gravity on the stability of the microtubes, fluid flows in from the left end and out the right end. The inner diameter and outer diameter of the microtubes are $R_{i}$ and $R_{o}$. The radius of the reference point is *r* and the length is *L*. Assume that the internal fluid is a steady flow, the velocity is *U*, the mass per unit length of the microtube is $m_{p}$ and the mass of the fluid per unit length is $m_{f}$. The transverse excitation load is $F\left(x\right)\cos\left(\omega t\right)$, where $F\left(x\right)=f\varphi_{1}\left(x\right)$. The structure of the functionally graded simply supported fluid-conveying microtubes is shown in Figure 1. The assumptions are as follows: (1) the fluid in the microtube is incompressible; (2) the flow of the internal fluid is a plunger flow; (3) the Euler–Bernoulli beam assumption is satisfied; (4) the movement of the microtubes is a plane movement; (5) functionally graded materials satisfy strain gradient theory; (6) the viscosity of the fluid is ignored.

The properties of functionally graded materials are distributed in accordance with the power law [37]. The material properties can be expressed as follows:
(1)$$E\left(r\right)=V_{i}E_{i}+V_{o}E_{o}$$
(2)$$G\left(r\right)=V_{i}G_{i}+V_{o}G_{o}$$
(3)$$\rho \left(r\right)=V_{i}\rho_{i}+V_{o}\rho_{o}$$
(4)$$V_{i}={\left(\frac{R_{o}-r}{R_{o}-R_{i}}\right)}^{n}$$
(5)$$V_{o}=1-V_{i}$$
where subscripts “*i*” and “*O*” refer to the inner layer and the outer layer of microtubes; *n* denotes the power-law index of volume fraction of the functionally graded materials; *r* is the radius of the reference point; *E*, *G* and *ρ* refer to the elastic modulus, shear modulus and densities of microtubes, respectively; *V* denotes the volume fraction of the functional gradient materials.

Based on the Euler–Bernoulli beam and strain gradient theory, the strain energy of the microtubes can be obtained [23]:
(6)$$U_{p}=\frac{1}{2}\int_{0}^{L}\left[S{\left(\frac{\partial^{2}w}{\partial x^{2}}\right)}^{2}+K{\left(\frac{\partial^{3}w}{\partial x^{3}}\right)}^{3}\right]dx$$
where $S=EI+2GAl_{0}^{2}+\frac{8}{15}GAl_{1}^{2}+GAl_{2}^{2}$ and $K=2GIl_{0}^{2}+\frac{4}{5}GIl_{1}^{2}$. *w* is the transversal displacement of microtubes, and $l_{0},l_{1},l_{2}$ represent the length scale parameter of functionally graded materials.

Ignoring the axial motion of the microtubes, the kinetic energy of the transversal vibration of the microtubes is as follows:
(7)$$T_{p}=\frac{1}{2}m_{p}\int_{0}^{L}{\left(\frac{\partial w}{\partial t}\right)}^{2}dx$$

The kinetic energy of the fluid in the microtubes is as follows:(8)$$T_{f}=\frac{1}{2}m_{f}\int_{0}^{L}{\left(\frac{\partial w}{\partial t}+U\frac{\partial w}{\partial x}\right)}^{2}dx$$

The work done by the transverse excitation loads on the microtubes is as follows [36]:(9)$$W_{F}=\int_{0}^{L}F(x)\cos(\omega t)dx$$
where *ω* is the frequency of transverse excitation loads.

The governing equation of the microtubes can be obtained by applying Hamilton’s principle as follows:
(10)$$\delta \int_{t_{1}}^{t_{2}}(T_{P}+T_{f}-U_{P})dt+\delta \int_{t_{1}}^{t_{2}}W_{F}dt=0$$

By substituting Equations (6)–(9) into Equation (10), the motion equation of the microtubes can be obtained as follows:
(11)$$\left(m_{p}+m_{f}\right)\frac{\partial^{2}w}{\partial t^{2}}+\left[m_{f}U^{2}-\frac{EA}{2L}\int_{0}^{L}{\left(\frac{\partial w}{\partial x}\right)}^{2}dx\right]\frac{\partial^{2}w}{\partial x^{2}}+2m_{f}U\frac{\partial^{2}w}{\partial x\partial t}+S\frac{\partial^{4}w}{\partial x^{4}}-K\frac{\partial^{6}w}{\partial x^{6}}=F\cos(\omega t)$$

For convenience of calculation, we introduce the following dimensionless quantities:
$$\begin{array}{c} \xi =\frac{x}{L},\;\eta =\frac{w}{L},\;\tau =\frac{t}{L^{2}}\sqrt{\frac{EI}{m_{f}+m_{p}}},\;u=\sqrt{\frac{m_{f}}{EI}}UL,\;\;\;\beta =\frac{m_{f}}{m_{p}+m_{f}},\;\kappa =\frac{2GIl_{2}^{2}}{EIL^{2}}\left(\frac{l_{0}^{2}}{l_{2}^{2}}+\frac{2l_{1}^{2}}{5l_{2}^{2}}\right),\\ \varepsilon =\frac{fL^{3}}{EI},\;\mu =\frac{EAL}{2EI},\;\Omega =\omega L^{2}\sqrt{\frac{m_{f}+m_{p}}{EI}},\;\;\;\Psi =\frac{GAl_{2}^{2}}{EI}\left(1+\frac{EI}{GAl_{2}^{2}}+\frac{2l_{0}^{2}}{l_{2}^{2}}+\frac{8l_{1}^{2}}{15l_{2}^{2}}\right) \end{array}$$

By substituting the dimensionless parameters above into Equation (11), the motion equation of the simply supported fluid-conveying microtubes can be written as:
(12)$$\frac{\partial^{2}\eta }{\partial \tau^{2}}+2u\sqrt{\beta }\frac{\partial^{2}\eta }{\partial \xi \partial \tau }+\left[u^{2}-\mu \int_{0}^{1}{\left(\frac{\partial \eta }{\partial \xi }\right)}^{2}d\xi \right]\frac{\partial^{2}\eta }{\partial \xi^{2}}+\psi \frac{\partial^{4}\eta }{\partial \xi^{4}}-\kappa \frac{\partial^{6}\eta }{\partial \xi^{6}}=\varepsilon \varphi_{1}(\xi )\cos(\Omega \tau )$$

## 3. Equation Solving

Equation (12) is a sixth-order nonlinear partial differential equation, which can be discretized into a second-order nonlinear ordinary differential equation by the Galerkin method. To this end, the transversal vibration displacement of the microtubes is written as follows:
(13)$$\eta \left(\xi ,\tau \right)=\sum_{i=1}^{N}\varphi_{i}\left(\xi \right)q_{i}\left(\tau \right)$$
where *N* is the order of the mode, $q_{i}\left(\tau \right)$ is the *i*-th mode coordinate; $\varphi_{i}\left(\xi \right)$ is the basis function of the *i*-th eigen-mode. For simply supported fluid-conveying microtubes, the basis function of the *i*-th eigen-mode is shown as follows:
(14)$$\varphi_{i}\left(\xi \right)=\sqrt{2}\sin\left(\lambda_{i}\xi \right)$$
where $\lambda_{i}$ is the *i*-th-order eigenvalue of simply supported beam. Equation (13) is substituted into Equation (12) multiply both sides of the equation by $\varphi_{j}\left(\xi \right)$, integrating with respect to *ξ* over the interval [0, 1] and making use of the orthogonal property of the characteristic function, the following equation can be obtained
(15)$$\begin{array}{c} \sum_{i,j=1}^{N}\varphi_{j}(\xi )\varphi_{i}(\xi )\frac{\partial^{2}q_{i}(\tau )}{\partial \tau^{2}}+2u\sqrt{\beta }\sum_{i,j=1}^{N}\varphi_{j}(\xi )\varphi_{i}^{\prime }(\xi )\frac{\partial q_{i}(\tau )}{\partial \tau }\\ +\left[u^{2}-\mu \int_{0}^{1}\left(\sum_{i=1}^{N}\frac{\partial \varphi_{i}(\xi )}{\partial \xi }q_{i}\left(\tau \right)\right)\right]\sum_{i,j=1}^{N}\varphi_{j}(\xi )\varphi_{i}^{\prime \prime }(\xi )q_{i}(\tau )\\ +\psi \sum_{i,j=1}^{N}\lambda_{r}^{4}\varphi_{j}(\xi )\varphi_{i}(\xi )q_{i}(\tau )-\kappa \sum_{i,j=1}^{N}\lambda_{i}^{4}\varphi_{j}(\xi )\frac{\partial^{2}\varphi_{i}(\xi )}{\partial \xi^{2}}q_{i}(\tau )=\sum_{j=1}^{N}\varepsilon \varphi_{1}\left(\xi \right)\varphi_{j}(\xi )\cos\left(\Omega \tau \right) \end{array}$$

Equation (15) can be written in matrix form as follows:
(16)$$M_{ij}\frac{\partial^{2}q_{i}}{\partial \tau^{2}}+C_{ij}\frac{\partial q_{i}}{\partial \tau }+K_{ij}q_{i}+K_{ij}^{n}{\left(q_{i}\right)}^{3}=P_{i}\cos\left(\Omega \tau \right)$$
where $q_{i}(\tau )$ is the population vector in generalized coordinates. The mass matrix $M_{i}{}_{j}$, the damping matrix $C_{ij}$ and the stiffness matrix $K_{ij}$ can be written as
$$\begin{array}{l} M_{ij}=\int_{0}^{1}\varphi_{i}(\xi )\varphi_{j}(\xi )d\xi ,\;\;\;C_{ij}=\gamma \int_{0}^{1}\varphi_{i}(\xi )\varphi_{j}(\xi )d\xi +2u\sqrt{\beta }\int_{0}^{1}{\varphi^{\prime }}_{i}(\xi )\varphi_{j}(\xi )d\xi \\ P_{i}=\int_{0}^{1}\varepsilon \varphi_{1}\left(\xi \right)\varphi_{i}\left(\xi \right)d\xi ,\;\;K_{ij}=\psi \omega_{i}^{4}\int_{0}^{1}\varphi_{i}(\xi )\varphi_{j}(\xi )d\xi +(u^{2}-\kappa \omega_{i}^{4})\int_{0}^{1}\varphi_{i}{{}^{\prime }}^{\prime }(\xi )\varphi_{j}(\xi )d\xi \\ K_{ij}^{n}=-\mu \left[\int_{0}^{1}{{\varphi^{\prime }}^{\prime }}_{i}\left(\xi \right)\varphi_{j}\left(\xi \right)d\xi \right]\left[\int_{0}^{1}{\left({\varphi^{\prime }}_{i}\left(\xi \right)\right)}^{2}d\xi \right] \end{array}$$

Equation (16) is the second-order nonlinear ordinary differential equation and is transformed into the first-order normal form. Based on the Runge–Kutta method, the motion trajectory and velocity of the microtubes are obtained for correlation analysis. The influence of each parameter on the dynamic properties of the system is studied.

## 4. Results and Discussions

The functionally graded simply supported fluid-conveying microtubes have an inner layer of zirconia ceramic and an outer layer of aluminum. Their material properties are *E_i_* = 151 GPa, *ρ_i_* = 3000 kg/m^3^, *E_o_* = 70 GPa, *ρ_o_* = 2707 kg/m^3^.

In this section, the length scale parameters of functionally graded materials are selected as $l=l_{0}=l_{1}=l_{2}=17.6\;\mu m$ [20]. Let the dimensionless microscale parameters be equal to the ratio of the outer diameter of the microtubes to the length scale parameter of the functionally graded materials, namely, *δ* = *D*/*l*. Taking the mass ratio *β* = 0.64 and the initial value of the system [0.000001, 0, 0, 0], the effects of functionally graded materials, dimensionless microscale parameters, excitation loads on microtube dynamic properties are studied.

### 4.1. Influence of Volume Fraction on Vibration Characteristics of the Microtubes

Functionally graded material is a new high-performance composite material, and its power-law index of volume fraction is an important indicator to characterize the composition of microtubes materials. A different power-law index corresponds to different material mechanical properties. Taking the dimensionless microscale parameter *δ* = 3, the dimensionless fluid velocity *u* = 4, and the power-law index of volume fraction, *n* = 0, 0.5, 5 and ∞, the motion trajectory phase diagram of microtubes is obtained, as shown in Figure 2.

According to Figure 2, as the power-law index of volume fraction increases, the vibration displacement and velocity of the microtubes increase. When *n* = 0, the microtube is a single homogeneous ceramic microtube with a maximum displacement is 0.00043. When *n* = 0.5, the maximum displacement of the microtubes is 0.00053. When *n* = 5, the maximum displacement of the microtubes is 0.00075. When *n* = ∞, the microtube is a single homogeneous metal microtube with a maximum displacement is 0.00085. In contrast, the maximum displacement of a single homogeneous metal microtube is 1.98 times that of a single homogeneous ceramic microtube. The reasons for the above changes are as follows. When *n* = 0, functionally graded materials degenerate into a single homogeneous zirconia ceramic. When *n* = ∞, functionally graded materials degenerate to a single homogeneous metal aluminum. Since the elastic modulus of aluminum is significantly lower than that of zirconia, the resistance to deformation is weakened. As the power-law index of the volume fraction increases, the stability of the microtubes decreases, and the vibration displacement increases.

### 4.2. Influence of Dimensionless Microscale Parameters on Vibration Characteristics of the Microtubes

Taking the power-law index of the volume fraction of functionally graded materials *n* = 1 and the dimensionless fluid velocity *u* = 6, the motion phase diagram of microtubes with different dimensionless microscale parameters is obtained, as shown in Figure 3. According to Figure 3, as the dimensionless microscale parameter increases, the microtube’s vibration displacement and velocity increase. When dimensionless microscale parameter *δ* = 1, the microtube performs quasi-periodic motion around the zero-equilibrium point. Despite the disordered motion trajectory, the orientation is similar, with a maximum displacement of the microtube is 0.000095. It can be seen from Figure 3b–d that the larger the dimensionless microscale parameter is, the more disordered the motion trajectory of the microtubes is.

However, these trajectories are not random, but with self-similar characteristics. Over time, any two trajectories will not coincide. The maximum displacements are 0.00089, 0.0013 and 0.0017, respectively. The displacement difference between Figure 3a,d is 17.89 times. The larger the dimensionless microscale parameter, the larger the diameter, the smaller the microscale effect and the worse the stability of microtubes.

### 4.3. Influence of Excitation Loads on Vibration Characteristics of the Microtubes

Taking the power-law index of the volume fraction *n* = 1, and the dimensionless microscale parameter *δ* = 3, displacement time history diagram and phase diagram of microtubes with different dimensionless amplitude of the excitation loads are obtained, as shown in Figure 4, Figure 5, Figure 6 and Figure 7.

According to Figure 4a and Figure 5a, when the dimensionless fluid velocity *u* = 1 and *u* = 3, the vibration displacement of the microtubes is proportional to the amplitude of excitation loads. The greater the amplitude of excitation loads, the greater the vibration displacement of the microtubes. As can be seen in Figure 4b and Figure 5b, the microtubes undergo quasi-periodic motion, which can also be verified in the time history diagram.

As can be seen from Figure 6, when the dimensionless fluid velocity *u* = 10; the time history diagram of the microtubes presents a disordered trajectory at the different amplitude of excitation loads. Over time, these trajectories do not coincide. This chaos phenomenon is also presented in Figure 7. In the chaos, the magnitude of the amplitude of excitation loads has an almost negligible effect on the magnitude of the microtube’s vibration displacement.

Taking the power-law index *n* = 1, and the amplitude of excitation loads *ε* = 0.003, the bifurcation diagram of the microtubes at different frequencies of excitation loads is obtained. The horizontal coordinate of the bifurcation diagram is the frequency of excitation loads, and the ordinate is the vibration displacement at the midpoint of the microtubes. When the vibration velocity at the midpoint of the microtubes is equal to 0, the instantaneous vibration displacement is recorded. Namely, when $\dot{\eta }\left(0.5,\tau \right)=0$, the approximate displacement at the midpoint of the microtubes $\eta \left(0.5,\tau \right)≃\varphi_{1}\left(0.5\right)q_{1}\left(\tau \right)+\varphi_{2}\left(0.5\right)q_{2}\left(\tau \right)$. Based on the above data, the microtube bifurcation diagram is shown in Figure 8.

As can be seen from Figure 8a, when the dimensionless fluid velocity *u* = 1 and the frequency of excitation loads Ω = 16.401, the vibration displacement of the microtubes increases sharply. Once the frequency of excitation loads deviates from this value, the vibration displacement of the microtubes will decrease significantly.

A similar phenomenon also appears in Figure 8b. the vibration displacement of the microtubes reaches its maximum when the frequency of excitation loads Ω = 10.591. According to Figure 8c, when the dimensionless fluid velocity *u* = 6, the microtubes undergo quasi-periodic motion. It can be seen from Figure 8d that the microtubes perform chaos when the dimensionless fluid velocity *u* = 8. The maximum value of the vibration displacement of the microtubes is significantly larger than the first three cases.

To further demonstrate the effect of the frequency of excitation loads on the dynamic properties of the microtubes, the displacement time history diagram of the microtubes with different dimensionless fluid velocities is shown in Figure 9, Figure 10, Figure 11 and Figure 12.

Figure 9 shows the displacement time history diagram of microtubes with different dimensionless frequencies of the excitation loads when the dimensionless velocity *u* = 1. As can be seen from Figure 9, the frequency of excitation loads has a very significant effect on the vibration characteristics of the microtubes. When the frequency of excitation loads Ω = 16.401, the dimensionless time of one vibration period is *τ* = 145, and the maximum displacement of the vibration is *η* = 0.0042. When the frequency of excitation loads deviates from this value, both the vibration displacement of the microtubes and the vibration time of one cycle will decrease significantly, as shown in Figure 9a,c. The dimensionless time of one vibration period is *τ* = 26 and *τ* = 24. The maximum value of vibration displacement is *η* = 0.00077 and *η* = 0.00069, respectively. Their one-period time difference is 5.58 times and 6.04 times, respectively. Furthermore, their maximum displacements differ by a factor of 5.45 times and 6.09 times. 

Figure 10 shows the displacement time history diagram of the microtubes with different dimensionless frequencies of the excitation loads when the dimensionless velocity *u* = 4, which has a similar law to that in Figure 9. As the dimensionless fluid velocity increases, the stability of the microtubes decreases and the vibration displacement increases. According to Figure 10b, when the frequency of excitation loads Ω = 10.599, the dimensionless time of one vibration period of microtubes is *τ* = 320, and the maximum displacement of the vibration is *η* = 0.0131. It can be seen from Figure 10a,c that the dimensionless time of one vibration period is *τ* = 42 and *τ* = 26, respectively. The maximum value of vibration displacement is *η* = 0.0018 and *η* = 0.0011, respectively. Their one-period time difference is 7.62 times and 12.31 times, respectively. Their maximum displacements differ by a factor of 7.28 times and 11.91 times.

According to the above analysis, when the microtubes are in a steady state, and the frequency of excitation loads is close to or equal to the natural frequency of the microtubes, the vibration of the microtubes will increase significantly. Once the frequency of excitation loads deviates from the microtube’s natural frequency, its effect on the microtubes will be significantly reduced. When the microtubes are unstable, the frequency of excitation loads will cause the instability of the fluid-conveying microtubes to increase, and the vibration displacement will increase significantly. In this case, chaotic motion of the microtube occurs, and the results are shown in Figure 11 and Figure 12.

## 5. Conclusions

In this paper, the nonlinear vibration characteristics of functionally graded simply supported fluid-conveying microtubes under transverse excitation loads are investigated. Based on the strain gradient theory and the Hamiltonian principle, the nonlinear motion equation of microtubes is derived. The motion trajectory and velocity of the microtubes are obtained by the Galerkin method and Runge–Kutta method. The effects of functionally graded materials, microscale parameters and excitation loads on the dynamic behavior of microtubes are explored. Some conclusions are obtained as follows:(1)Changes in the power-law index of the volume fraction of functionally graded materials affect the effective bending stiffness of the materials. By choosing an appropriate power-law index, the stability of the microtubes can be improved. The larger the dimensionless microscale parameters, the worse the stability of the microtubes and the more disordered the motion trajectory.(2)The amplitude of excitation loads has a significant influence on the dynamic behavior of the microtubes. When the microtubes are in a steady state, the greater the amplitude of excitation loads, the greater the vibration displacement of the microtubes. Once the microtubes are unstable, the effect of the amplitude of excitation loads on the vibration displacement of the microtubes is almost negligible.(3)The frequency of excitation loads leads to microtube resonance. When the microtubes are in a steady state, and the frequency of excitation loads is close to or equal to the natural frequency of the microtubes, the vibration displacement of the microtubes increases significantly. Once the frequency of excitation loads deviates from the microtube’s natural frequency, its effect on the microtubes will be significantly reduced. When the microtubes are in an unstable state, the frequency of excitation loads will cause the instability of the microtubes to increase, and the vibration displacement will increase significantly, resulting in chaos for the microtubes.(4)The influence of transverse excitation loads on the stability of fluid-conveying microtubes is very great, especially the frequency of excitation loads. Therefore, the application of microtubes should avoid the effect of excitation loads.

## Figures and Tables

**Figure 1 micromachines-13-02114-f001:**
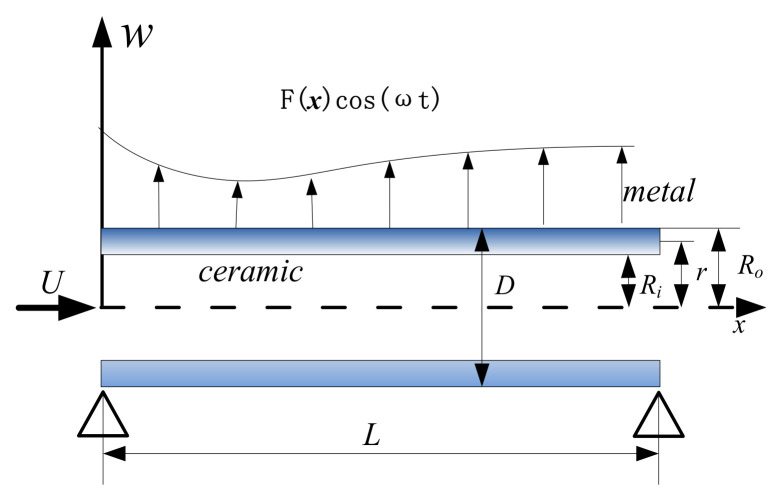
Structure diagram of functionally graded simply supported fluid-conveying microtubes.

**Figure 2 micromachines-13-02114-f002:**
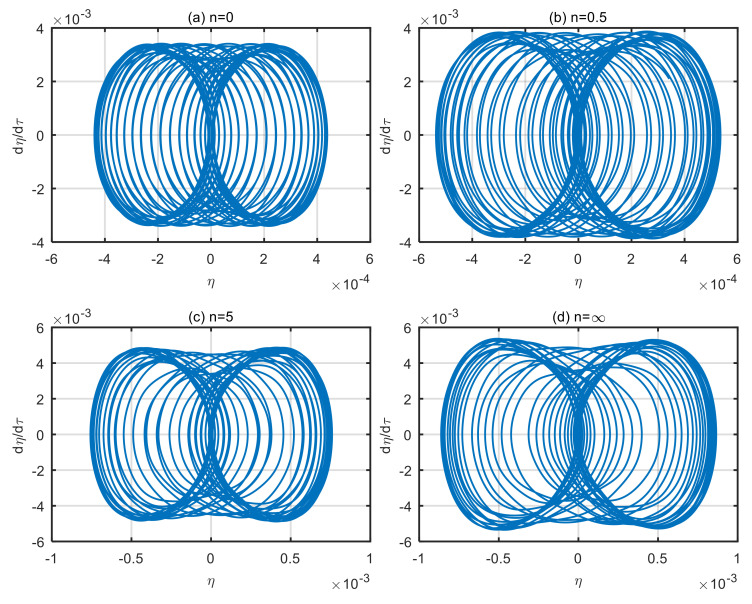
Phase diagram of microtubes with different power-law index of the volume fraction. (**a**) *n* = 0; (**b**) *n* = 0.5; (**c**) *n* = 5; (**d**) *n* = ∞.

**Figure 3 micromachines-13-02114-f003:**
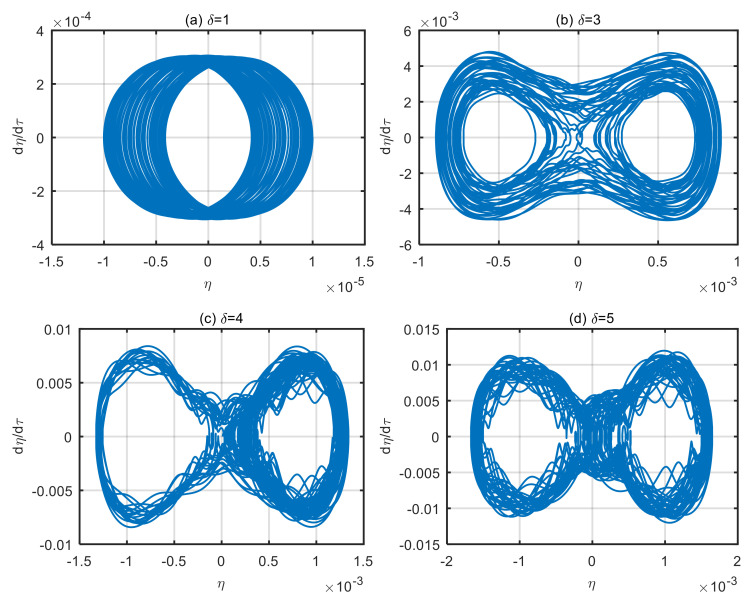
Motion phase diagram of the system with different dimensionless microscale parameters. (**a**) *δ* = 1; (**b**) *δ* = 3; (**c**) *δ* = 4; (**d**) *δ* = 5.

**Figure 4 micromachines-13-02114-f004:**
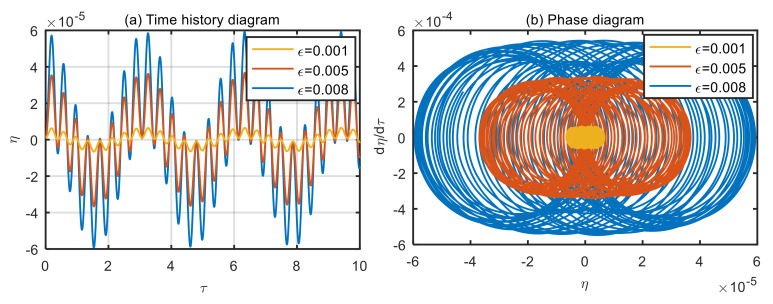
Displacement time history diagram (**a**) and phase diagram (**b**) of microtubes with different dimensionless amplitude of the excitation loads (*u* = 1).

**Figure 5 micromachines-13-02114-f005:**
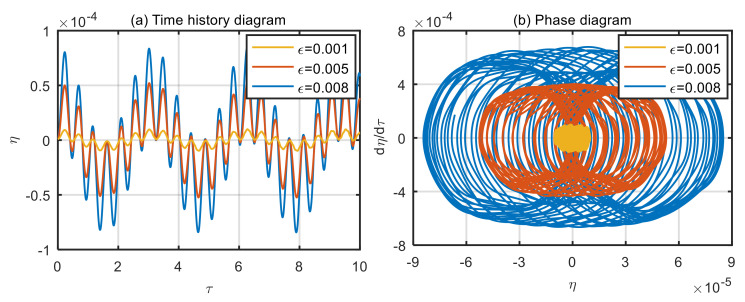
Displacement time history diagram (**a**) and phase diagram (**b**) of microtubes with different dimensionless amplitude of the excitation loads (*u* = 3).

**Figure 6 micromachines-13-02114-f006:**
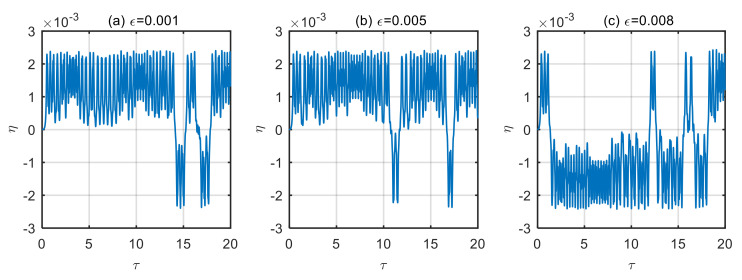
Displacement time history diagram of microtubes with different dimensionless amplitude of the excitation loads (*u* = 10). (**a**) *ε* = 0.001; (**b**) *ε* = 0.005; (**c**) *ε* = 0.008.

**Figure 7 micromachines-13-02114-f007:**
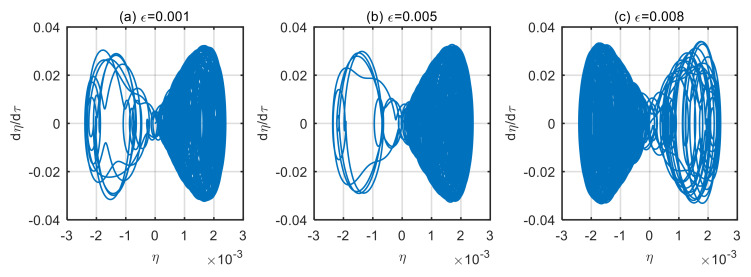
Motion phase diagram of microtubes with different dimensionless amplitude of the excitation loads (*u* = 10). (**a**) *ε* = 0.001; (**b**) *ε* = 0.005; (**c**) *ε* = 0.008.

**Figure 8 micromachines-13-02114-f008:**
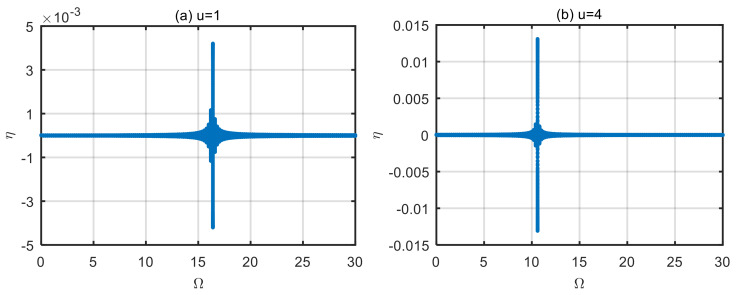
Bifurcation diagram of microtube displacement with dimensionless frequency of the excitation loads. (**a**) *u* = 1; (**b**) *u* = 4; (**c**) *u* = 6; (**d**) *u* = 8.

**Figure 9 micromachines-13-02114-f009:**
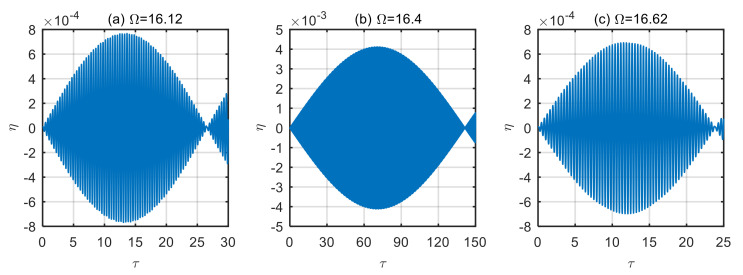
Displacement time history diagram of microtubes with different dimensionless frequencies of excitation loads (*u* = 1). (**a**) Ω = 16.12; (**b**) Ω = 16.4; (**c**) Ω = 16.62.

**Figure 10 micromachines-13-02114-f010:**
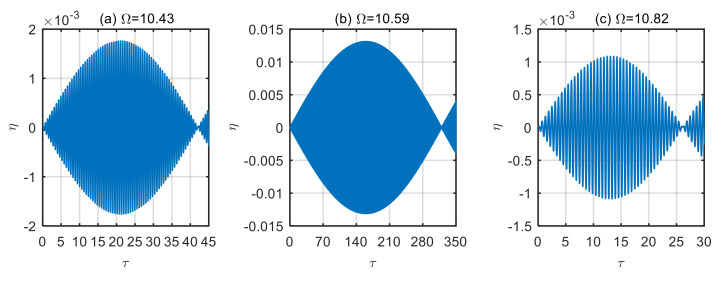
Displacement time history diagram of microtubes with different dimensionless frequencies of excitation loads (*u* = 4). (**a**) Ω = 10.43; (**b**) Ω = 10.59; (**c**) Ω = 10.82.

**Figure 11 micromachines-13-02114-f011:**
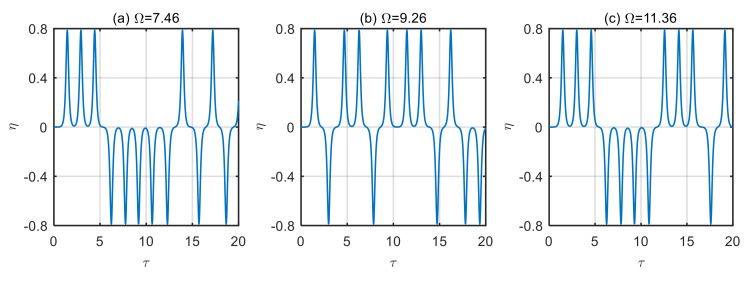
Displacement time history diagram of microtubes with different dimensionless frequencies of excitation loads (*u* = 6). (**a**) Ω = 7.46; (**b**) Ω = 9.26; (**c**) Ω = 11.36.

**Figure 12 micromachines-13-02114-f012:**
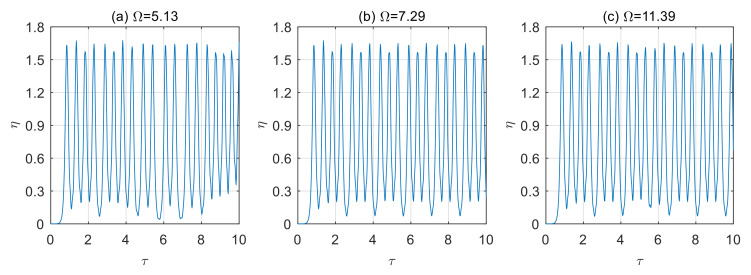
Displacement time history diagram of microtubes with different dimensionless frequencies of excitation loads (*u* = 8). (**a**) Ω = 5.13; (**b**) Ω = 7.29; (**c**) Ω = 11.39.

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
