# Peer review of "Analysis of Nonlinear Vibration of Functionally Graded Simply Supported Fluid-Conveying Microtubes Subjected to Transverse Excitation Loads"

_micromachines, 2022, doi:10.3390/mi13122114_

Round 1

Reviewer 1 Report

Forced vibration of graded fluid-conveying microscale tubes under an external harmonic load is studied. I recommend this article if the authors consider the following major comments:

1- What is the meaning of phi_1 in line 143? Is this different from the term in equation 13?

2- The parameters which have been investigated in the article are limited and their effect on the dynamic behavior of the system is obvious. Therefore, the innovation of the article is weak.

3- The literature review is not complete. Sufficient explanations have not been given about the different methods of analysis and control of vibration as well as modeling of actual small-scale systems. Use the following articles in the introduction section: DOI: 10.1007/s11071-022-07243-7. DOI: 10.1115/1.4054324. DOI: 10.1016/j.bspc.2020.102367.

4- The results of the article are stated without proper validation.

Author Response

Point 1: What is the meaning of phi_1 in line 143? Is this different from the term in equation 13?

Response 1: Thank you very much for your comments. The phi_1 in line 143 represents the variation law of the excitation load amplitude along the axial direction of the microtubes. The phi_1 in the Eq. (13) represents the basis function of the i-th eigen-mode. when i=1, the phi_1 in the Eq. (13) is the same as the phi_1 in line 143.

Point 2: The parameters which have been investigated in the article are limited and their effect on the dynamic behavior of the system is obvious. Therefore, the innovation of the article is weak.

Response 2: your comments are very good. Many parameters can affect the dynamic behavior of the microtubes. For example, the effects of size effect, thermal load and partially distributed tangential force on the dynamic behavior of the carbon nanotubes were studied in Reference [2], [5] and [28] respectively. This paper mainly analyzes the dynamic behavior of functionally graded fluid-conveying microtubes under transverse excitation loads, so the main research focus is on excitation loads frequency and amplitude.

Point 3:  The literature review is not complete. Sufficient explanations have not been given about the different methods of analysis and control of vibration as well as modeling of actual small-scale systems. Use the following articles in the introduction section: DOI: 10.1007/s11071-022-07243-7. DOI: 10.1115/1.4054324. DOI: 10.1016/j.bspc.2020.102367.

Response 3: Thank you very much for your opinions. We think your opinions is very helpful to us, so we quoted the following literature, DOI: 10.1007/s11071-022-07243-7.

Point 4: The results of the article are stated without proper validation.

 Response 4: Your suggestions are very good. Experimental verification is extremely important, however, due to the lack of experimental conditions for micro and nano structures, we are not able to compare and analyze current studies with specific experimental results, which is also our biggest issue.

Reviewer 2 Report

Journal: Journal of Micromachines

Manuscript ID: micromachines-2061301

 Analysis of Nonlinear Vibration of Functionally Graded Simply Supported Fluid-Conveying Microtubes Subjected to Transverse Excitation Loads

The paper has investigated nonlinear dynamics of FG microtubes under transverse excitation loads with internal fluid flow. Based on the Euler-Bernoulli theory, Hamilton principle and strain gradient theory, the nonlinear equations of motion were derived and the solved by Galerkin method. The dynamical characteristics of FG microtubes including natural frequencies, time responses and chaos were analyzed in the paper. Overall, the paper is written well. However, before making the final decision, the following revision should be addressed carefully:

1.    The Introduction of the paper is poorly written, disproportionate to the quality of the paper. The writing style is quite sketchy and lists previous works. The authors need to polish and refresh the Introduction and deeply emphasize the meaning of the problem, new points and practical applications. In addition, some FG structures should also be included in the literature review: European Journal of Mechanics-A/Solids, 86, 104168, 2021; Thin-Walled Structures, 159, 107204, 2021; AIAA Journal, 59, 1, 366-378, 2021; Thin-Walled Structures, 146, 106414, 2020. Some papers related to fluid-structure interaction should be reviewed carefully: Aerospace Science and Technology, 92, 501-519, 2019; Composite Structures, 162, 164-181, 2017; Engineering Structures, 198, 109502.

2.    What theory does the strain equations use here? Please specify. The conveying-fluid part is not given in sections 2 and 3. So how is the velocity U of the flow represented?

3.    Figure 1 does not clearly show the problem model. The authors should separate into 2 figures: a material model and a structural model.

4.    The results of the paper are quite interesting; however, the analysis is too sketchy and lacks appreciation of the significance of the physical and mechanical aspects. The authors need to evaluate further and, if necessary, can provide Poincare map to illustrate the dynamical responses in periodic and chaotic states.

5.    There were some errors in grammar and typing (missing articles a, an and the; error formula ...). The conclusions need to clarify the new point of the article instead of listing the phenomena of the problem results.

Author Response

Point 1: The Introduction of the paper is poorly written, disproportionate to the quality of the paper. The writing style is quite sketchy and lists previous works. The authors need to polish and refresh the Introduction and deeply emphasize the meaning of the problem, new points and practical applications. In addition, some FG structures should also be included in the literature review: European Journal of Mechanics-A/Solids, 86, 104168, 2021; Thin-Walled Structures, 159, 107204, 2021; AIAA Journal, 59, 1, 366-378, 2021; Thin-Walled Structures, 146, 106414, 2020. Some papers related to fluid-structure interaction should be reviewed carefully: Aerospace Science and Technology, 92, 501-519, 2019; Composite Structures, 162, 164-181, 2017; Engineering Structures, 198, 109502.

Response 1: Thank you very much for your opinions. We think your opinions is very helpful to us, so we quoted the following two literatures:Thin-Walled Structures, 146, 106414, 2020. Aerospace Science and Technology, 92, 501-519, 2019.

Point 2: What theory does the strain equations use here? Please specify. The conveying-fluid part is not given in sections 2 and 3. So how is the velocity U of the flow represented?

Response 2: your comments are very good. The strain equations were derived based on the stain gradient theory. The stain gradient theories were developed by Lam (Experiments and theory in strain gradient elasticity. Journal of the Mechanics and Physics of Solids, 2003, 51, 1477–1508), which three length scale parameters appear in the constitutive equation. Based on the strain gradient theory, Ref [20] studied the size dependent nonlinear dynamic analysis of functionally graded conveying-fluid microtubes.

U denotes the flow velocity. The fluid flow in the microtubes is non-viscous, incompressible, and irrotational. It is explained in the section 2.

Point 3: Figure 1 does not clearly show the problem model. The authors should separate into 2 figures: a material model and a structural model.

Response 3: This is a very good suggestion. According to your suggestions, we have added a structure drawing, and hope it can meet your expectations.

Point 4: The results of the paper are quite interesting; however, the analysis is too sketchy and lacks appreciation of the significance of the physical and mechanical aspects. The authors need to evaluate further and, if necessary, can provide Poincare map to illustrate the dynamical responses in periodic and chaotic states.

Response 4: Thank you very much for your comments. Poincare diagram is a good way to describe the nonlinear dynamic behavior of structures. Then, for the analysis in this paper, the time-history diagram and phase plane diagram have been able to well show the nonlinear dynamic behavior of the excitation load amplitude and frequency on the microtubule.

Point 5: There were some errors in grammar and typing (missing articles a, an  and the; error formula ...). The conclusions need to clarify the new point of the article instead of listing the phenomena of the problem results.

Response 5: We are very sorry for our incorrect writing. We have checked the whole article and made corrections. According to your suggestions, we have also made some modification and marked them in red in the conclusions.

Round 2

Reviewer 1 Report

All the requested modifications have been made. The article is acceptable as it is and can be published in its current form.

Reviewer 2 Report

The paper is significantly improved and it can be considered for publication.